# Greywater Reuse: Contaminant Profile, Health Implications, and Sustainable Solutions

**DOI:** 10.3390/ijerph22050740

**Published:** 2025-05-07

**Authors:** Phumudzo Budeli, Linda Lunga Sibali

**Affiliations:** Department of Environmental Sciences, College of Agriculture and Environmental Sciences, University of South Africa, P.O. Box 392, Florida 1710, South Africa

**Keywords:** greywater reuse, water scarcity, pollutants, health risks, endocrine disrupting chemicals (EDCs), microbial contaminants

## Abstract

Global water scarcity is becoming an increasingly critical issue; greywater reuse presents a promising solution to alleviate pressure on freshwater resources, particularly in arid and water-scarce regions. Greywater typically sourced from household activities such as laundry, bathing, and dishwashing, constitutes a significant portion of domestic wastewater. However, the reuse of greywater raises concerns about the potential risks posed by its complex composition. Despite the growing body of literature on greywater reuse, most studies only focus on specific contaminants, thus there is a limited understanding of the comprehensive profile of contaminants, health, and environmental effects associated with these pollutants. This review adds new knowledge through a holistic exploration of the composition and physico-chemical characteristics of greywater, with a focus on its organic and inorganic pollutants, heavy metals, EDCs, emerging microplastics, nanoparticles, and microbial agents such as bacteria, fungi, viruses, and protozoa. This review sheds light on the current state of knowledge regarding greywater pollutants and their associated risks while highlighting the importance of safe reuse. Additionally, this review highlights the removal of contaminants from greywater and the sustainable use of grey water for addressing water scarcity in affected regions.

## 1. Introduction

Water scarcity is an escalating global concern, driven by population growth, urbanization, and climate change. As freshwater resources become increasingly limited, greywater reuse has emerged as a potential solution to alleviate pressure on water supplies, particularly in arid and water-scarce regions [1]. Greywater, derived from non-toilet household activities such as bathing, laundry, and dishwashing, constitutes a significant proportion of domestic wastewater. It is reused for non-potable purposes such as irrigation and toilet flushing and offers a promising strategy for reducing freshwater demand and promoting sustainable water management [2]. However, while greywater reuse holds considerable potential, it also presents substantial challenges due to its complex and variable composition. The composition of greywater reflects its diverse sources, containing a mixture of organic and inorganic substances, heavy metals, endocrine-disrupting chemicals (EDCs), and microbial contaminants [3].

The presence of these pollutants raises concerns about potential health and environmental risks associated with untreated or poorly treated greywater [4]. Organic matter and nutrients in greywater can contribute to eutrophication when discharged into the environment, while heavy metals and EDCs may pose long-term toxicological effects on human health and ecosystems [5]. Additionally, microbial contaminants such as bacteria, viruses, fungi, and protozoa can cause waterborne diseases, further complicating efforts to promote greywater reuse [6,7]. Despite the growing practice of greywater reuse in many regions, the understanding of its health and environmental impacts remains limited, emphasizing the need for comprehensive assessment and effective management strategies [8].

The review provides a detailed analysis of these contaminants and their potential health risks, highlighting the importance of robust treatment and filtration systems to ensure the safe reuse of greywater [9]. Effective greywater treatment technologies are essential to mitigate these risks, ensuring that greywater reuse can be a viable and sustainable solution for addressing water scarcity without compromising public health and environmental integrity [10,11]. Through synthesizing current knowledge on greywater composition and associated risks, this review underscores the importance of adopting safe and sustainable greywater reuse practices. It also identifies critical research gaps and practical challenges that must be addressed to promote widespread and responsible implementation.

## 2. Greywater Composition and Characteristics

The physico-chemical characteristics of greywater play a crucial role in determining the selection and effectiveness of treatment methods [12]. These characteristics, including FOG, pH, temperature, suspended solids, organic matter content, nutrients, and presence of contaminants such as surfactants and heavy metals, influence the behavior of pollutants and the performance of treatment processes [13,14]. Evaluating these parameters is essential in designing sustainable and efficient treatment systems for greywater.

### 2.1. Fats, Oil and Grease (Fog)

The FOG presents significant challenges in wastewater management due to its detrimental effects on water quality [15]. Their accumulation forms an oily layer on the water surface, impeding light penetration, oxygen diffusion, and the process of photosynthesis by submerged plants. This, in turn, disrupts the ecological balance of aquatic ecosystems [16]. In untreated household wastewater, FOG concentrations typically range from 50 to 100 mg/L, posing a considerable threat to water bodies and treatment facilities alike [17]. Moreover, FOG impedes the functionality of filtration units, thereby reducing the efficiency of wastewater treatment processes [18].

Domestic grey water sources contribute to FOG pollution, with kitchen sinks and bathroom showers being major contributors [17]. Conventional methods for treating oily wastewater encompass a range of techniques such as chemical treatment, biological flotation, gravity flotation methods, dissolved air flotation (DAF), and membrane processes [19]. Extensive research has been conducted on the efficacy of gravity separation and dissolved air flotation in removing FOG from wastewater, achieving removal efficiencies of up to 85% for emulsified oils [20]. However, the natural treatment of wastewater containing fats and oils, while beneficial, is often inadequate when dealing with dispersed forms of FOG [21]. A holistic approach incorporating proactive measures at the source, such as heightened public awareness campaigns and regulatory interventions promoting eco-friendly practices and alternatives, is essential for curbing FOG pollution and safeguarding the ecological integrity of aquatic environments for posterity.

### 2.2. Temperature

Temperature fluctuations have a significant impact on microbial activity and treatment efficiency in greywater systems, influencing both the rate of microbial growth and metabolism as well as the performance of biological treatment processes [22]. Understanding these effects is crucial for optimizing greywater treatment strategies and ensuring water quality in sustainable water management practices [11]. Microbial activity in greywater treatment systems is highly temperature dependent. Temperature affects the growth rate, reproduction, and metabolic processes of microorganisms present in greywater [23]. Generally, microbial activity increases with higher temperatures, as warmer conditions provide an optimal environment for microbial growth and enzymatic reactions [11]. Conversely, lower temperatures can inhibit microbial activity, leading to reduced treatment efficiency and longer retention times required for effective treatment [23].

As depicted in Table 1, according to [24], the temperature in kitchen greywater ranged between 24.4 °C and 30.0 °C, while bathroom greywater exhibited a slightly narrower range of 25.8 °C to 29.0 °C. These values are particularly significant as they fall within the mesophilic range (20–40 °C), which is known to support microbial activity in biological treatment systems. However, the implications of this thermal profile on greywater treatment performance remain an area of ongoing research, particularly in the context of decentralized, low-energy systems suited for household or peri-urban reuse. Studies such as [25] suggests that the treatment efficiency of biological systems—such as constructed wetlands, sequencing batch reactors (SBR), and aerobic biofilters—is enhanced at temperatures above 20 °C, given that microbial enzymatic reactions are more active within this range. In particular, heterotrophic bacteria responsible for the degradation of organic matter (BOD and COD) demonstrate optimal growth around 30 °C, which correlates well with the temperature ranges observed by [26]. This suggests that greywater from kitchens and bathrooms in warmer climates may inherently support more efficient organic pollutant removal when subjected to biological treatment.

In anaerobic digestion, elevated temperatures enhance the activity of anaerobic bacteria, facilitating the breakdown of complex organic compounds into simpler forms such as methane and carbon dioxide [32]. Temperature fluctuations can also influence the composition and diversity of microbial communities in greywater systems [33]. Certain microorganisms may be more resilient to temperature variations, leading to shifts in microbial populations and potentially altering treatment performance [34,35]. Additionally, extreme temperature conditions, such as thermal shocks, can disrupt microbial activity and compromise treatment efficiency in greywater systems [36].

Optimizing greywater treatment strategies to account for temperature fluctuations is essential for ensuring consistent treatment performance and water quality [37]. Strategies may include implementing temperature control measures, such as insulation or heat exchange systems, to maintain stable temperatures within treatment units [37]. Furthermore, selecting and acclimating microbial consortia adapted to local temperature conditions can enhance treatment resilience and efficiency in greywater systems [38]. Therefore, understanding the effects of temperature on microbial growth, metabolism, and treatment processes is essential for optimizing greywater treatment strategies and ensuring the sustainable management of water resources.

### 2.3. pH Levels

Among the factors affecting greywater treatment, pH levels play a crucial role in influencing pollutant solubility and treatment processes [22]. The pH of greywater significantly affects the solubility of pollutants present in the wastewater. For example, at higher pH levels (alkaline conditions), certain pollutants such as phosphates and detergents tend to be more soluble, potentially leading to increased nutrient levels in treated effluent [1]. Conversely, under acidic conditions (lower pH levels), some metals and organic compounds may become more soluble, posing challenges for treatment processes [39].

The pH of greywater significantly influences the selection and performance of treatment technologies, with reported values varying across sources and usage zones as shown in Table 1. Kitchen greywater tends to exhibit greater variability and acidity, ranging from 5.5 ± 0.5 ([27]; [40] 6.5–7.7 [28], 5.58–10.0 [24], and 5.9–7.4 [41]). In contrast, bathroom greywater is typically near neutral to slightly alkaline, with values between 7.3–7.8 [30,31], 7.1–7.6 [28], and 5.98–8.40 [24]. Treatment processes are sensitive to these pH variations: biological systems, such as aerobic filters or constructed wetlands, generally perform optimally within a near-neutral range (6.5–8.0), as microbial activity is inhibited at more acidic or highly alkaline conditions. Acidic kitchen greywater (pH < 6) may require pre-neutralization or buffering to maintain microbial health and ensure enzymatic activity in biodegradation processes [41]. Furthermore, chemical treatment methods like coagulation-flocculation and advanced oxidation processes (AOPs) are also pH-dependent, with coagulation efficiency often peaking at pH 6–7 and oxidative degradation (e.g., with ozone or H_2_O_2_) being more effective in alkaline conditions. Membrane systems are less pH-sensitive structurally but may experience fouling or reduced flux when pH alters the solubility of contaminants or affects charge interactions. Therefore, the wide pH range observed in greywater highlights the need for tailored pretreatment strategies to stabilize pH before the main treatment stages, ensuring consistent and efficient contaminant removal across different greywater sources.

### 2.4. Total Suspended Solids (TSS)

The characteristics of suspended solids in greywater are essential for selecting appropriate treatment strategies to ensure effective pollutant removal and environmental sustainability [42]. Suspended solids can adversely influence water quality and treatment processes, necessitating thorough evaluation and appropriate treatment measures [42]. Greywater typically contains a range of suspended solids with varying particle sizes and concentrations. These solids can include organic and inorganic particles, colloids, and microorganisms [43]. The concentration of suspended solids in greywater varies depending on factors such as the source of wastewater, household activities, and the effectiveness of pre-treatment measures [44].

Total Suspended Solids (TSS) in greywater exhibit substantial variability depending on the source, with kitchen greywater generally showing significantly higher concentrations than bathroom greywater. Kitchen greywater has been reported to contain TSS levels ranging from 11 to 3934 mg/L [24] and 134 to 1300 mg/L [41], primarily due to the presence of food residues, oils, and grease. In contrast, bathroom greywater demonstrates comparatively lower TSS levels, ranging from 58 to 78 mg/L [28] and 19 to 793 mg/L [24], largely attributable to the dilution of solids in water from showers, handwashing, and personal hygiene activities as shown in Table 1. These disparities have direct implications for treatment: high TSS in kitchen greywater demands robust pre-treatment methods, such as grease traps, sedimentation tanks, or coarse filtration, to prevent clogging and fouling in subsequent biological or membrane systems. Biological treatment processes like constructed wetlands or biofilters are effective when TSS is reduced to manageable levels (<300 mg/L), as excessive solids can inhibit microbial activity by reducing oxygen transfer or causing physical smothering of biofilms. In contrast, bathroom greywater with moderate TSS can often bypass intensive pre-treatment, requiring only basic filtration or sedimentation before entering biological units. The wide range in TSS values underscores the necessity of source-specific treatment designs, emphasizing modular or adaptable systems that can accommodate fluctuating solid loads, particularly in decentralized or household-level greywater reuse schemes.

### 2.5. Chemical Oxygen Demand (COD)

Chemical oxygen demand (COD) is a key parameter used to assess organic pollution in greywater, which acts as an indicator of organic pollution and a relevant parameter for selecting and optimizing treatment strategies [45]. COD levels are crucial for designing effective treatment systems to mitigate organic pollution and ensure environmental sustainability [46]. These organic compounds contribute to the pollution load of greywater and can adversely influence water quality and ecosystem health. Monitoring COD levels provides valuable insight into the organic pollution content of greywater and informs treatment decisions aimed at pollutant removal [47]. COD is a measure of the amount of oxygen required to chemically oxidize organic and inorganic compounds in water [48]. In greywater, elevated COD levels indicate a higher concentration of organic pollutants, including biodegradable organic matter, detergents, and other contaminants. High COD levels in greywater signify increased organic pollution, which can lead to oxygen depletion in receiving water bodies and adversely affect aquatic ecosystems [49,50].

COD in greywater is a critical parameter indicating the organic load and the potential demand for oxygen during treatment, with values varying significantly between kitchen and bathroom sources. Kitchen greywater, characterized by high organic input from food particles, oils, and detergents, shows markedly elevated COD levels ranging from 770–2050 mg/L [27,40], 58–1340 mg/L [24], and even up to 8071 mg/L in extreme cases [41]. Similarly, [28] reported COD concentrations as low as 110 ± 100 mg/L, highlighting the heterogeneity of household practices and water usage. In contrast, bathroom greywater tends to have lower COD concentrations, typically within the range of 64–903 mg/L [30,31], with [24,28] noting values around 575 ± 98 mg/L and 230–367 mg/L, respectively. The treatment of greywater with such variable COD levels necessitates tailored approaches: low to moderate COD (<500 mg/L) can often be managed using natural treatment systems like constructed wetlands, sand filters, or sequencing batch reactors, which offer cost-effective and sustainable solutions. However, higher COD levels, particularly those exceeding 1000 mg/L, require more intensive interventions, such as aerobic biological treatment. Elevated COD also correlates with an increased risk of microbial proliferation, unpleasant odors, and reduced treatment efficiency if not adequately pre-treated. These findings further highlight the need for source separation strategies and staged treatment processes to effectively handle greywater of varying strength, ensuring both treatment efficiency and the safe reuse of water.

### 2.6. Biochemical Oxygen Demand (BOD)

One of the key parameters used to assess organic pollution in water is BOD [51]. Greywater composition varies depending on the source and activities within households. Common sources of greywater include showers, sinks, laundry machines, and dishwashers [24]. Each source contributes different types and amounts of organic matter to the greywater stream. For example, laundry wastewater may contain higher levels of organic compounds from detergents and fabrics, while water from showers may have lower BOD levels due to the absence of detergent residues [45]. Consequently, BOD levels in greywater can fluctuate widely based on the household’s activities and practices [3,52].

Organic content is a major contributor to BOD levels in greywater. Organic matter in greywater primarily includes substances such as food particles, oils, soaps, detergents, and other biodegradable materials [53]. These organic compounds serve as substrates for microbial metabolism, driving the biochemical processes that consume oxygen and contribute to BOD [54,55]. Therefore, higher levels of organic content in greywater lead to elevated BOD levels. The correlation between organic content and BOD underscores the importance of understanding and managing the sources of organic pollution in greywater [56]. Effective treatment of greywater is essential to reduce BOD levels and mitigate its environmental impact.

Kitchen greywater typically shows significantly higher BOD levels due to its higher organic load from food particles, oils, fats, and detergents. Ref. [28] reported BOD levels ranging from 40.8–890 mg/L, while [24] observed a broader and more elevated range between 185–2460 mg/L, and [28] recorded values from 536–1460 mg/L. These elevated concentrations imply that kitchen greywater can exert substantial oxygen demand in receiving environments if not properly treated, potentially leading to oxygen depletion and degradation of aquatic ecosystems. Conversely, bathroom greywater largely derived from bathing, handwashing, and laundry typically presents lower BOD concentrations, though still requiring adequate treatment. Refs. [30,31] reported average BOD values around 166 ± 37 mg/L, while [28] noted a range of 129–173 mg/L, and [24] reported 20–673 mg/L. These values, though generally lower than those for kitchen sources, still exceed the safe discharge and reuse thresholds without treatment. Treatment technologies must therefore be tailored to the BOD loading. For lower ranges typical of bathroom greywater, natural systems like reed beds, sand filtration, and biofilters are often sufficient. However, for higher BOD levels, especially in kitchen effluent, more robust biological treatments such as sequencing batch reactors (SBRs), aerobic digestion, or hybrid systems incorporating membrane or chemical oxidation steps are necessary. High BOD not only challenges treatment efficiency but can also lead to odor problems and system overloading. Thus, understanding the BOD profile in greywater is essential for optimizing treatment system design and ensuring the safe and sustainable reuse of this resource.

## 3. Presence of Heavy Metals and Their Dynamics in Greywater

Heavy metals can be present in greywater due to various sources such as household cleaning products, metal pipes, and corrosion of plumbing fixtures [52]. These metals, including lead, cadmium, mercury, chromium, and zinc, pose potential risks to human health and the environment if not properly managed [57]. The occurrence of heavy metals in greywater is of concern due to their toxicity and persistence in the environment [57]. Heavy metals can accumulate in soils, water bodies, and organisms, leading to bioaccumulation and bio-magnification in the food chain [58]. Human exposure to heavy metals through contaminated water or food can result in adverse health effects such as neurological disorders, kidney damage, and carcinogenicity [59].

### 3.1. Lead, Cadmium, Mercury

Heavy metals are frequently present in grey wastewater due to various human activities, including, household practices, and urban runoff [60]. Among the heavy metals commonly found in grey wastewater, lead (Pb), cadmium (Cd), and mercury (Hg) are of particular concern due to their toxic properties and potential risks to human health and the environment [61,62]. These heavy metals enter grey wastewater through multiple pathways, including, atmospheric deposition, and domestic sources such as plumbing materials, consumer products, and household waste [63].

Lead is a highly toxic heavy metal that can cause severe health effects, particularly in children and pregnant women [57]. Exposure to lead through grey wastewater can occur via the corrosion of lead-containing plumbing materials in domestic water distribution systems, leading to elevated lead concentrations in drinking water and wastewater [64,65]. Chronic exposure to lead can impair neurological development, cognitive function, and cardiovascular health, while acute exposure can cause symptoms such as abdominal pain, nausea, and renal dysfunction [66]. Lead contamination in grey wastewater poses risks of environmental pollution and human exposure, highlighting the importance of mitigating sources of lead contamination in water systems [67,68].

Cadmium is another toxic heavy metal commonly found in grey water, primarily originating from urban runoff, and household waste [69]. Cadmium is known for its carcinogenic and nephrotoxic properties, posing risks of adverse health effects such as kidney damage, bone demineralization, and reproductive toxicity [70,71]. Chronic exposure to cadmium through contaminated water sources can lead to long-term health impacts, particularly in populations with higher susceptibility such as children, elderly individuals, and individuals with compromised renal function [71]. Cadmium contamination in grey wastewater underscores the importance of monitoring and urban runoff to mitigate environmental pollution and human health risks [72].

Mercury is a highly toxic heavy metal that exists in various forms, including elemental mercury, inorganic mercury compounds, and organic mercury compounds such as methylmercury [73,74]. Mercury contamination in grey water can arise from activities such as coal combustion, mining operations, and chemical manufacturing, as well as domestic sources such as thermometers, fluorescent lamps, and consumer products [75]. Mercury poses significant risks to human health and the environment, particularly through the bioaccumulation and biomagnification of methylmercury in aquatic food chains [76]. Chronic exposure to methylmercury can lead to neurological disorders, developmental impairments, and cardiovascular effects, posing risks to human populations dependent on contaminated aquatic ecosystems for food and livelihoods [77]. Mitigating sources of heavy metal contamination in grey wastewater is essential for protecting human health and the environment, highlighting the importance of monitoring and urban runoff to minimize environmental pollution and mitigate human exposure risks [78].

### 3.2. Copper and Zinc

Copper and zinc are two essential trace elements that are commonly found in grey wastewater, originating from various sources including household plumbing materials and consumer products [79]. These metals are often present in grey wastewater due to their widespread use in plumbing fixtures, water pipes, and household appliances, as well as their inclusion in consumer products such as detergents, cosmetics, and personal care items [25,80]. Copper and zinc are essential micronutrients for humans, playing critical roles in enzyme function, metabolism, and immune system function [81]. However, elevated levels of copper and zinc in grey wastewater can have adverse effects on ecosystems and human health [82]. The sources of copper and zinc in grey wastewater vary depending on human activities and household practices [83]. Copper is commonly used in plumbing materials, water distribution systems, and household appliances due to its corrosion resistance and antimicrobial properties [84]. Zinc is widely used in galvanized steel, brass fittings, and consumer products such as batteries, paints, and cosmetics [85]. These metals enter grey wastewater through corrosion of plumbing materials, leaching from consumer products, contributing to their presence in domestic wastewater systems [86].

Elevated levels of copper and zinc in grey water can have adverse effects on aquatic ecosystems and human health. In aquatic environments, copper and zinc can accumulate in sediments, surface waters, and aquatic organisms, leading to toxicity and ecological disruptions [87]. Copper and zinc are known to exert toxic effects on aquatic organisms such as fish, invertebrates, and algae, affecting growth, reproduction, and survival [88]. Additionally, copper and zinc can interfere with biochemical processes in aquatic organisms, leading to impaired enzyme function, oxidative stress, and cellular damage [89,90]. Human exposure to copper and zinc in grey wastewater can occur through various pathways, including ingestion of contaminated water, dermal contact, and consumption of contaminated food and aquatic organisms [91]. Chronic exposure to elevated levels of copper and zinc can lead to adverse health effects such as gastrointestinal disturbances, neurological symptoms, and liver and kidney damage [92]. Vulnerable populations such as children, pregnant women, and individuals with underlying health conditions may be particularly susceptible to the adverse effects of copper and zinc exposure [93].

### 3.3. Chromium and Nickel

Chromium and nickel are two heavy metals frequently found in grey wastewater, primarily originating from household practices [94]. Chromium is commonly used in industrial processes such as electroplating, stainless steel production, and leather tanning, while nickel is used in various applications including metal plating, alloy production, and electronics manufacturing [95]. These metals can enter grey wastewater through atmospheric deposition, and domestic sources such as consumer products, plumbing materials, and household waste [86,96]. The industrial sources of chromium and nickel in grey water pose significant challenges for wastewater treatment due to the complex nature of their chemical forms and their resistance to conventional treatment methods [97]. Chromium exists in various oxidation states, including trivalent chromium (Cr (III)) and hexavalent chromium (Cr (VI)), with Cr (VI) being more toxic and mobile in aqueous environments [98]. Nickel is present in multiple chemical forms, including soluble nickel salts and insoluble nickel oxides, which can undergo redox reactions and complexation processes in wastewater systems [99]. These complex chemical forms of chromium and nickel can pose challenges for their removal and treatment in wastewater treatment plants [100].

The treatment of chromium and nickel in grey wastewater is challenging due to their resistance to conventional treatment processes and the limited effectiveness of existing treatment technologies [94]. Conventional wastewater treatment processes such as sedimentation, filtration, and biological treatment are generally ineffective in removing chromium and nickel from wastewater due to their complex chemical forms and low solubility [101]. Advanced treatment technologies such as membrane filtration, ion exchange, and chemical precipitation may be required to effectively remove chromium and nickel from grey wastewater, but these methods are often costly and may require specialized equipment and expertise [53,102]. In addition to treatment challenges, the occurrence of chromium and nickel in grey wastewater raises concerns about their potential environmental and human health impacts [2]. Chromium and nickel can accumulate in sediments, surface waters, and aquatic organisms, leading to toxicity and ecological disruptions in aquatic ecosystems [103]. Chronic exposure to elevated levels of chromium and nickel in contaminated water sources can pose risks to human health, including respiratory effects, gastrointestinal disturbances, and carcinogenic effects [104]. Table 2 shows that different heavy metals have been detected in different parts of the globe with concentrations that are hazardous to humans, wildlife, and the environment, this further affirms the need to treat grey wastewater before reuse.

## 4. Presence of Organic Contaminants in Grey Wastewater

### 4.1. Surfactants and Detergents

#### 4.1.1. Anionic Surfactants

Anionic surfactants are commonly found in grey wastewater due to their extensive use in cleaning agents and household detergents [108]. These surfactants, characterized by their negatively charged hydrophilic (water-attracting) head and hydrophobic (water-repelling) tail, are effective in reducing surface tension and facilitating the removal of dirt, grease, and other organic materials during cleaning activities [109]. Common types of anionic surfactants include alkyl sulfates, alkyl ether sulfates, and linear alkylbenzene sulfonates, which are widely used in laundry detergents, dishwashing liquids, and household cleaners due to their excellent foaming and cleaning properties [110]. As a result of their extensive use in household cleaning products, anionic surfactants are frequently discharged into domestic wastewater systems, contributing to their presence in grey wastewater [111].

The presence of anionic surfactants in grey wastewater raises concerns about their potential environmental impacts, particularly in aquatic ecosystems receiving treated effluent [112]. Anionic surfactants can persist in aquatic environments due to their low biodegradability and resistance to degradation by microbial communities [113]. These surfactants can accumulate in sediments, surface waters, and aquatic organisms, posing risks to aquatic life and ecosystem health [111]. Anionic surfactants can disrupt aquatic ecosystems by altering surface water tension, affecting the behavior and physiology of aquatic organisms such as fish, amphibians, and invertebrates [114]. Additionally, anionic surfactants may contribute to the formation of foam in surface waters, impairing gas exchange processes and affecting the ecological balance of aquatic ecosystems [115].

The environmental fate and behavior of anionic surfactants in grey wastewater are influenced by factors such as surfactant type, concentration, wastewater treatment processes, and receiving water body characteristics [108].Conventional wastewater treatment processes such as sedimentation, filtration, and biological treatment are generally effective in removing suspended solids and organic matter, including anionic surfactants, from wastewater [94]. However, some anionic surfactants may resist degradation or sorb onto sludge particles during treatment, leading to their persistence in treated effluent and potential discharge into receiving water bodies [116]. Advanced treatment technologies such as membrane filtration, activated carbon adsorption, and ozonation may enhance the removal of anionic surfactants from wastewater, reducing their environmental impact [117].

#### 4.1.2. Cationic Surfactants

Cationic surfactants are a class of chemicals commonly found in grey wastewater, originating from various sources including household cleaning products, personal care, and items [118]. These surfactants contain positively charged hydrophilic (water-attracting) heads and hydrophobic (water-repelling) tails, allowing them to effectively bind to negatively charged surfaces and facilitate the removal of dirt and grease during cleaning activities [109]. Common types of cationic surfactants include alkyltrimethylammonium compounds (e.g., cetrimonium bromide), alkylbenzyldimethylammonium compounds (e.g., benzalkonium chloride), and quaternary ammonium compounds (QACs), which are widely used in household disinfectants, fabric softeners, and personal care products due to their antimicrobial properties and surfactant capabilities [119].

The occurrence of cationic surfactants in grey wastewater can have significant effects on water quality, particularly in terms of aquatic toxicity and environmental persistence [111]. Cationic surfactants are known for their antimicrobial properties, which can contribute to the inhibition of microbial activity in wastewater treatment processes and receiving water bodies [120]. However, excessive concentrations of cationic surfactants in wastewater effluent may lead to adverse effects on aquatic organisms and ecosystems, including toxicity to fish, invertebrates, and algae [110]. Cationic surfactants can disrupt the integrity of cell membranes and interfere with cellular processes in aquatic organisms, leading to physiological stress, reduced growth, and impaired reproductive success [119].

Domestic sources of cationic surfactants include the use of household cleaning products, disinfectants, fabric softeners, and personal care items containing these compounds [111]. The release of cationic surfactants into grey wastewater can occur through direct disposal down drains, washing machines, and other household appliances, as well as through indirect pathways such as surface runoff and atmospheric deposition [121].

#### 4.1.3. Non-Ionic Surfactants

Non-ionic surfactants are a class of chemicals commonly found in grey wastewater, characterized by their lack of electrical charge in the hydrophilic (water-attracting) head group [122]. These surfactants are composed of hydrophilic head groups, typically polyethylene oxide or polypropylene oxide chains, and hydrophobic (water-repelling) tail groups, often consisting of alkyl or aryl chains [123]. Non-ionic surfactants are widely used in household cleaning products, laundry detergents, and personal care items due to their versatility, compatibility with various water conditions, and low foam-forming properties [124]. Common types of non-ionic surfactants include alcohol ethoxylates, alkylphenol ethoxylates, and fatty acid ethoxylates, which are added to cleaning formulations to enhance wetting, emulsification, and soil removal capabilities [125].

The behavior of non-ionic surfactants in grey wastewater during treatment processes depends on various factors, including surfactant properties, wastewater composition, treatment methods, and environmental conditions [126]. Non-ionic surfactants are generally less toxic and more environmentally friendly compared to cationic and anionic surfactants, owing to their neutral charge and lower tendency to interact with biological membranes [22]. These surfactants exhibit good biodegradability under aerobic conditions, with microbial communities in wastewater treatment plants capable of metabolizing and degrading non-ionic surfactants through enzymatic reactions [22,113].

During wastewater treatment processes, non-ionic surfactants undergo physical, chemical, and biological transformations that contribute to their removal and degradation [127]. Physical processes such as sedimentation, filtration, and adsorption can facilitate the removal of non-ionic surfactants by trapping them onto solid particles or sorbing them onto activated carbon or other adsorbents [111]. Chemical processes such as oxidation and photolysis may also contribute to the degradation of non-ionic surfactants, particularly under advanced treatment conditions such as ozonation, UV irradiation, and advanced oxidation processes [128].

Biological degradation of non-ionic surfactants by microbial communities in wastewater treatment plants plays a significant role in their removal from grey wastewater [108]. Microorganisms such as bacteria, fungi, and archaea possess enzymatic capabilities to metabolize and degrade non-ionic surfactants through enzymatic reactions such as hydrolysis, oxidation, and conjugation [129]. Non-ionic surfactants are generally more biodegradable compared to anionic and cationic surfactants, with microbial communities capable of utilizing these compounds as carbon and energy sources for growth and metabolism [22]. However, the efficiency of biological degradation of non-ionic surfactants in wastewater treatment systems may vary depending on factors such as compound structure, concentration, and treatment conditions [130]. The behavior of cationic, ionic, and non-ionic surfactants in grey wastewater during treatment processes is influenced by their properties, wastewater composition, treatment methods, and environmental conditions. There is a need to study the fate and behavior of non-ionic surfactants in grey wastewater to assess their environmental impact and implement effective wastewater management and treatment strategies to mitigate their release and minimize their adverse effects on aquatic ecosystems

### 4.2. Emerging Chemical Contaminants

#### 4.2.1. Pharmaceuticals and Personal Care Products (PPCPs)

Pharmaceuticals and personal care products (PPCPs) are a diverse group of chemicals that are commonly found in grey wastewater due to their widespread use in human activities [131]. PPCPs encompass a wide range of compounds, including prescription and over-the-counter medications, personal care products such as fragrances and cosmetics, and household chemicals like disinfectants and cleaning agents as depicted in Figure 1 [132]. These compounds can enter grey wastewater through various routes, including excretion by humans and animals, disposal of unused medications, and washing off personal care products during bathing and cleaning activities [45]. As a result, grey wastewater serves as a significant pathway for the release of PPCPs into the aquatic environment [133].

The fate and degradation of PPCPs in grey wastewater are influenced by various factors, including the physicochemical properties of the compounds, wastewater treatment processes, and the activity of microbial communities [134]. PPCPs exhibit diverse physicochemical properties, including solubility, hydrophobicity, and stability, which influence their behavior in wastewater systems [135]. Some PPCPs may undergo transformation or degradation during wastewater treatment processes, while others may persist and remain detectable in treated effluent and receiving water bodies [133]. Microbial communities present in grey wastewater play a crucial role in the fate and degradation of PPCPs through biotransformation, biodegradation, and sorption processes [136].

Microbial communities in grey wastewater comprise diverse populations of bacteria, fungi, archaea, and viruses, which possess enzymatic capabilities to metabolize and degrade PPCPs through various metabolic pathways [137,138]. Bacteria are particularly important for the degradation of organic compounds, including PPCPs, through enzymatic reactions such as hydroxylation, oxidation, reduction, and conjugation [139]. Fungi and archaea also contribute to PPCP degradation through enzymatic activities, although their roles may vary depending on the specific compounds and environmental conditions [140]. Additionally, microbial communities can interact with PPCPs through sorption onto microbial biomass or extracellular polymeric substances (EPS), affecting the bioavailability and fate of PPCPs in wastewater systems [141,142].

The degradation and removal of PPCPs by microbial communities in wastewater treatment processes depend on factors such as microbial diversity, activity, and environmental conditions [143]. Conventional wastewater treatment processes such as activated sludge, trickling filters, and biological reactors utilize microbial communities to degrade organic matter and remove pollutants, including PPCPs, from wastewater [94,108]. Advanced treatment technologies such as membrane bioreactors, ozonation, and UV irradiation may enhance the removal of PPCPs by targeting specific compounds or improving overall treatment efficiency [144]. However, the efficiency of microbial degradation of PPCPs in wastewater treatment systems may vary depending on factors such as compound properties, microbial activity, and treatment conditions [143,145].

#### 4.2.2. Endocrine Disruptors in Grey Wastewater

Endocrine-disrupting chemicals (EDCs) are a group of diverse compounds commonly found in grey wastewater, originating from various sources including industrial processes, consumer products, and agricultural practices [146,147]. EDCs are chemicals that interfere with the endocrine system by mimicking, blocking, or disrupting the action of hormones in organisms, leading to adverse effects on development, reproduction, metabolism, and other physiological processes [148]. The presence of EDCs in grey wastewater is attributed to their use in a wide range of applications, including plastics, pesticides, flame retardants, personal care products, and pharmaceuticals, as well as their incomplete metabolism and excretion by humans and animals [46].

The occurrence of EDCs in grey wastewater raises concerns about their potential impacts on aquatic ecosystems and human health [149]. EDCs can enter aquatic environments through wastewater discharges, surface runoff, and atmospheric deposition, leading to contamination of surface waters, sediments, and aquatic organisms [146,150]. Once released into the environment, EDCs can bioaccumulate in aquatic organisms and biomagnify through food chains, posing risks of toxicity and ecological disruptions to aquatic ecosystems [151]. Chronic exposure to EDCs in contaminated water sources can lead to adverse health effects in aquatic organisms, including altered reproductive function, developmental abnormalities, endocrine disruption, and impaired immune function [152,153].

In addition to their impacts on aquatic ecosystems, EDCs in grey wastewater can pose risks to human health through various exposure pathways [147]. Human exposure to EDCs can occur through ingestion of contaminated water and food, dermal contact, inhalation of contaminated air, and direct contact with consumer products containing EDCs [154,155]. Chronic exposure to EDCs has been associated with various health effects in humans, including reproductive disorders, developmental abnormalities, metabolic disorders, immune dysfunction, and increased risk of hormone-related cancers [146,156]. Vulnerable populations such as pregnant women, infants, and children may be particularly susceptible to the adverse effects of EDC exposure due to their developmental stages and increased sensitivity to hormone disruptions as shown in Figure 1 [157,158]. The presence of EDCs in grey wastewater raises concerns about their potential impacts on aquatic ecosystems and human health, including toxicity to aquatic organisms and adverse health effects in humans.

#### 4.2.3. Nanoparticles and Microplastics in Grey Water

The detection of nanoparticles (NPs) and microplastics (MPs) in greywater has emerged as a significant concern in environmental science and water reuse policy. These emerging contaminants are often overlooked in conventional greywater treatment paradigms, yet they pose substantial challenges to ecological safety and human health [159]. The fact that they are ubiquitous, persistent, and largely unregulated status in many national guidelines makes them particularly problematic for the sustainable reuse of greywater in both agricultural and non-potable domestic applications [159]. MPs are defined as plastic particles less than 5 mm in size and are increasingly detected in greywater sources, especially from laundry effluents where synthetic textiles release vast quantities of microfibres during washing cycles [160]. These particles are chemically stable and hydrophobic, making them potential vectors for the adsorption and transport of other pollutants, including heavy metals, pesticides, and pathogenic microorganisms [161]. Their persistence in the environment is further compounded by the inability of most conventional treatment systems such as sand filters or sedimentation units to effectively capture or degrade them. Consequently, MPs often enter soil and aquatic systems when greywater is reused for irrigation, potentially disrupting soil microbial communities and entering food chains through plant uptake or trophic transfer [160].

Some engineered nanoparticles, such as silver, zinc oxide, and titanium dioxide, are commonly introduced into domestic greywater through personal care products, sunscreens, detergents, and antimicrobial agents [162]. Nanoparticles exhibit high reactivity and mobility due to their small size and large surface area, enabling them to interact with biological membranes and induce oxidative stress in living organisms [163]. Although some nanoparticles possess antimicrobial properties, their uncontrolled release through untreated or partially treated greywater poses ecotoxicological risks, particularly in soil and aquatic environments where they can alter nutrient cycles and microbial diversity [163]. Moreover, the long-term implications of chronic low-level exposure to NPs via greywater reuse remain poorly understood, warranting further toxicological studies [164].

Despite growing awareness, the monitoring and regulation of NPs and MPs in greywater remain limited, primarily due to analytical challenges in detection and quantification, as well as the absence of standardized treatment protocols targeting these contaminants [165]. Technologies such as advanced oxidation processes (AOPs), membrane filtration (especially nanofiltration and ultrafiltration), and biochar adsorption have shown promise in experimental settings. Other aspects such as their scalability, energy requirements, and economic feasibility in decentralized greywater reuse systems are still under debate [166]. A critical gap in the literature is the lack of risk assessment frameworks that incorporate emerging contaminants like MPs and NPs into reuse safety guidelines. Current reuse practices often focus on traditional pollutants such as BOD, COD, and fecal indicators while less attention is paid to the potential long-term impacts of micropollutants. Addressing these challenges requires interdisciplinary research combining environmental toxicology, material science, and wastewater engineering to develop effective treatment strategies and regulatory policies.

### 4.3. Microbial Communities and Their Pathogenesis in Grey Wastewater

#### 4.3.1. Bacterial Communities and Their Pathogenesis

Bacterial pathogens are commonly found in grey wastewater, originating from various sources such as human fecal contamination, environmental reservoirs, and sewage discharge [167]. Grey wastewater, which includes domestic wastewater from household activities excluding toilet flushing, can serve as a reservoir for pathogenic bacteria shed from human feces and other sources [168]. These bacterial pathogens include a wide range of species such as *Escherichia coli*, *Salmonella* spp., *Campylobacter* spp., *Enterococcus* spp., and other enteric bacteria commonly associated with gastrointestinal illnesses [169,170]. Moreover, environmental sources such as soil, water, and animals can contribute to the presence of bacterial pathogens in grey wastewater through various routes of contamination [98].

The survival and persistence of bacterial pathogens in grey wastewater depend on factors such as temperature, pH, nutrient availability, and the presence of competing microorganisms [171,172]. Bacterial pathogens shed from human feces can survive and remain viable in grey wastewater for extended periods under favorable environmental conditions [173]. Some bacterial pathogens, particularly those with robust stress response mechanisms, may exhibit greater resistance to environmental stressors and disinfection treatments commonly used in wastewater treatment processes [174]. Furthermore, the presence of organic matter and nutrients in grey wastewater can provide substrates for bacterial growth and proliferation, enhancing the survival of pathogenic bacteria and their potential for transmission to humans and the environment [40].

The presence of bacterial pathogens in grey wastewater has significant implications for human health, particularly in scenarios where untreated or inadequately treated wastewater is discharged into water bodies used for drinking water supply and recreational activities [175,176]. Pathogenic bacteria in grey wastewater can pose risks of waterborne diseases and infections, including gastrointestinal illnesses, respiratory infections, and skin infections, particularly in vulnerable populations such as children, the elderly, and immunocompromised individuals [177]. Inadequate treatment or disposal of grey wastewater can lead to microbial contamination of water sources, posing public health risks and necessitating interventions to ensure safe water supply and sanitation [178].

#### 4.3.2. Viral Communities and Their Pathogenesis

The presence of viruses in grey wastewater poses substantial risks to both public health and the environment, necessitating thorough consideration in wastewater management practices [165,179]. These viruses, including enteric viruses such as norovirus, rotavirus, adenovirus, and hepatitis A virus, have the potential to cause waterborne diseases and infections in humans [180]. Moreover, they can persist in grey wastewater and pose risks to environmental health through various pathways [181,182].

The public health risks associated with viral contamination in grey wastewater are significant, particularly in scenarios where untreated or inadequately treated wastewater is discharged into water bodies used for drinking water supply or recreational activities [183]. Viral pathogens in grey wastewater are known to cause gastrointestinal illnesses, respiratory infections, and other diseases, with vulnerable populations such as children, the elderly, and immunocompromised individuals being particularly at risk [184]. Furthermore, environmental contamination by viruses in grey wastewater can disrupt aquatic ecosystems and pose risks to wildlife and ecosystem health [185]. Viral contamination through irrigation with untreated grey wastewater also presents concerns for soil quality and crop safety, potentially leading to foodborne viral infections in humans [186].

The persistence and resistance of viruses in grey wastewater vary depending on factors such as viral type, environmental conditions, and treatment processes [187]. Some enteric viruses, such as norovirus, are known to be highly resistant to environmental stressors and can persist in wastewater for extended periods, posing challenges to effective viral removal during treatment [188]. Additionally, certain viruses may exhibit resistance to disinfection methods commonly used in wastewater treatment, necessitating advanced treatment technologies to ensure viral inactivation and removal [189]. Effective monitoring and control measures are essential for managing viral contamination in grey wastewater and minimizing associated public health and environmental risks [190]. Routine monitoring of viral indicators in wastewater provides valuable insights into viral prevalence, persistence, and removal efficiency during treatment processes [191]. The implementation of robust treatment systems, incorporating physical, chemical, and biological processes, is necessary to achieve adequate viral removal and disinfection [84]. Furthermore, public education, regulatory oversight, and adherence to wastewater management guidelines are critical for preventing viral contamination and safeguarding human health and environmental sustainability in the context of grey wastewater management [192].

#### 4.3.3. Fungal Communities and Their Pathogenesis

The presence and diversity of fungi in grey wastewater have been documented, highlighting their potential impact on water quality and their diverse applications in various fields [193]. Fungi are ubiquitous microorganisms found in natural and engineered environments, including wastewater treatment systems [194]. Studies have reported the presence of fungal species belonging to diverse taxa such as Ascomycota, Basidiomycota, and Zygomycota in grey wastewater samples collected from different sources [195,196]. These fungi may originate from human activities, organic matter decomposition, or environmental sources and can thrive in the nutrient-rich and moist conditions of wastewater environments [197]. The presence of fungi in grey wastewater can affect water quality through various mechanisms, including nutrient cycling, organic matter degradation, and pathogen control [198]. Fungi play essential roles in nutrient recycling by decomposing organic matter and releasing nutrients such as nitrogen and phosphorus into the water [199]. Moreover, certain fungal species possess enzymatic capabilities that enable them to degrade complex organic compounds, contributing to wastewater treatment and bioremediation processes [200]. However, some fungi may also pose risks to water quality by producing mycotoxins or causing fungal contamination in water distribution systems, particularly in untreated or poorly treated wastewater [201]. The diversity and dynamics of fungal communities in grey wastewater are essential for assessing their impact on water quality and implementing effective treatment strategies [202,203].

Beyond their implications for water quality, fungi in grey wastewater have potential applications in various fields, including biotechnology and environmental engineering [204]. Fungal species isolated from wastewater environments have been explored for their biotechnological potential in producing enzymes, biofuels, and biopolymers through fermentation processes [205,206]. Additionally, certain fungi have been investigated for their role in wastewater treatment and bioremediation applications, including the removal of pollutants such as heavy metals, organic contaminants, and pharmaceuticals through fungal-mediated processes [207]. Fungi also play a beneficial role as biofertilizers, biocontrol agents against plant pathogens, and promoters of plant growth and nutrient uptake [208,209].

#### 4.3.4. Microbial Parasites and Their Pathogenesis

Parasites are important contributors to the microbial community present in grey wastewater, adding to the complexity of microbial contamination and posing significant risks to public health [210]. The parasitic pathogens include protozoa such as *Giardia lamblia, Cryptosporidium* spp., *Entamoeba histolytica*, and *helminths* such as *Ascaris lumbricoides*, *Trichuris trichiura*, and hookworms [211,212]. The presence of parasites in grey wastewater is particularly concerning due to their potential to cause waterborne diseases and infections in humans through ingestion, inhalation, or dermal contact with contaminated water [4]. Parasitic pathogens in grey wastewater can pose significant risks to public health, particularly in scenarios where untreated or inadequately treated wastewater is discharged into water bodies used for drinking water supply and recreational activities [6,213]. Protozoan parasites such as *Giardia* and *Cryptosporidium* are known to cause gastrointestinal illnesses, including diarrhea, abdominal cramps, and nausea, particularly in vulnerable populations such as children, the elderly, and immunocompromised individuals [214]. Helminthic parasites such as *Ascaris* and *Trichuris* can cause intestinal infections and parasitic diseases, leading to malnutrition, anemia, and impaired physical and cognitive development, particularly in resource-limited settings [215]. The presence of parasitic pathogens in grey wastewater underscores the importance of effective wastewater treatment to mitigate the risks of parasitic contamination and protect public health [216].

The treatment processes vary in their effectiveness for removing parasitic pathogens from grey wastewater, with some methods being more successful than others [217]. Conventional treatment processes such as sedimentation, filtration, and disinfection are commonly used in wastewater treatment plants to remove suspended solids and inactivate microbial pathogens, including parasites [218,219]. However, the efficacy of these processes for parasitic removal may vary depending on factors such as parasite species, size, and resistance to environmental stressors [220]. Advanced treatment technologies such as membrane filtration, UV irradiation, ozonation, and chlorine dioxide treatment have been shown to achieve higher levels of parasitic inactivation and removal in wastewater [221]. These technologies offer promising solutions for enhancing the safety and quality of treated wastewater effluent and minimizing the risks of parasitic contamination [222,223]. Implementation of advanced treatment technologies and rigorous monitoring protocols are crucial for ensuring the safety and quality of treated wastewater effluent in wastewater treatment plants.

## 5. Benefits, Challenges, and Sustainable Use of Grey Water

The potential benefits of using untreated greywater are often weighed against the associated risks to public health and environmental sustainability [10]. One of the primary advantages of utilizing untreated greywater is its potential to reduce freshwater consumption in non-potable applications such as irrigation and toilet flushing [224]. Studies have shown that untreated greywater, when appropriately managed, can serve as a valuable resource for other non-drinking water purposes and agricultural purposes, contributing to water conservation efforts [2]. The use of untreated greywater may alleviate the burden on wastewater treatment systems, particularly in areas facing water scarcity or lacking adequate infrastructure for conventional treatment processes [11]. Furthermore, untreated greywater often contains nutrients such as nitrogen and phosphorus, which can serve as fertilizers for plants when applied judiciously. This inherent nutrient content can enhance soil fertility and promote plant growth, thus contributing to sustainable agricultural practices [11]. Moreover, the relatively lower organic load of untreated greywater compared to black water facilitates its easier handling and application in various non-potable reuse scenarios, potentially reducing the energy and cost requirements associated with conventional wastewater treatment [45].

Despite its potential benefits, the utilization of untreated greywater poses significant challenges, primarily concerning public health risks and environmental contamination [225]. Untreated greywater may contain a plethora of contaminants, including pathogens, heavy metals, organic pollutants, and pharmaceutical residues, which can pose risks to human health and ecosystem integrity if not adequately managed [226]. Exposure to untreated greywater, particularly through direct contact or aerosolization, can lead to the transmission of waterborne diseases and pose risks to both workers and end-users [2]. Moreover, the indiscriminate use of untreated greywater without proper monitoring and regulation may result in environmental degradation, including soil and groundwater contamination, surface water pollution, and ecosystem disruption [183].

The presence of contaminants such as surfactants, endocrine-disrupting chemicals (EDCs), and pharmaceuticals in untreated greywater raises concerns about their potential impacts on aquatic organisms and ecosystem dynamics [227]. Additionally, the accumulation of nutrients from untreated greywater in receiving water bodies can lead to eutrophication, exacerbating water quality issues and compromising ecosystem health [47].

Previous reviews of existing technologies highlight several trade-offs between cost, treatment efficiency, space requirements, and maintenance needs. While decentralized greywater treatment systems offer flexibility and water-saving benefits, many existing solutions are either too technologically complex or unaffordable for widespread use in low- and middle-income urban or peri-urban households [228]. Furthermore, inconsistent water quality standards and a lack of regulatory frameworks for non-potable water reuse further limit the implementation of greywater systems in many developing countries [229]. Public acceptance and awareness also play a pivotal role, as negative perceptions around water aesthetics and hygiene can hinder the adoption of sustainable greywater reuse technologies [230]. A significant knowledge gap exists in the long-term performance, user acceptability, and environmental impact of greywater systems under real-world conditions. Many studies rely on controlled lab-scale experiments with limited replication in diverse household or climatic settings. Moreover, there is a lack of integrated systems that balance high contaminant removal efficiency with affordability, low energy consumption, and ease of use.

## 6. Conclusions

Greywater reuse has garnered significant attention as a viable alternative water source, particularly in regions experiencing acute water scarcity. However, the complex and variable composition of greywater necessitates a cautious and informed approach to its reuse. Recent studies highlight the importance of thoroughly understanding the physico-chemical properties of greywater, which often include elevated levels of biochemical oxygen demand (BOD), chemical oxygen demand (COD), total suspended solids (TSS), and temperatures that may facilitate microbial proliferation. While heavy metals are typically reported in low or non-detectable concentrations, their potential accumulation and associated toxicities cannot be overlooked. Moreover, the presence of emerging contaminants such as EDCs, pharmaceuticals, microplastics, and nanoparticles even at trace concentrations poses long-term ecological and public health concerns. Microbial agents, including bacteria, viruses, protozoa, and fungi, further complicate the risk profile of untreated greywater, as indicator organisms often exceed safe thresholds. Despite these challenges, evidence suggests that with appropriate and robust treatment technologies, greywater can be safely reclaimed for non-potable applications such as irrigation and toilet flushing. Therefore, the development and implementation of effective treatment strategies are critical to mitigating the associated risks and realizing the potential of greywater reuse as a sustainable water management solution. While considerable progress has been made in understanding and improving greywater treatment, challenges remain in bridging the gap between laboratory innovations and practical, scalable solutions. A multidisciplinary approach combining technical innovation with socio-economic, environmental, and policy considerations is essential for the safe and sustainable use of greywater in domestic and commercial settings. Future research should prioritize the development of low-cost, modular, and user-friendly greywater treatment technologies that are adaptable to various socio-economic and environmental contexts. Greater focus is also needed on policy development, community-based approaches, and education to promote safe and sustainable greywater reuse. Furthermore, research into nature-based solutions, hybrid systems that combine biological and physico-chemical processes, and digital water quality monitoring tools such as IoT-integrated sensors may offer pathways for more resilient and efficient greywater reuse systems.

## Figures and Tables

**Figure 1 ijerph-22-00740-f001:**
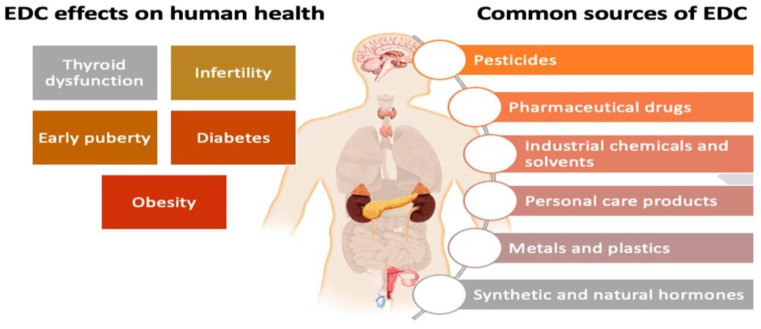
Health effects of EDCs in grey wastewater modified from [8].

**Table 1 ijerph-22-00740-t001:** A table of some physico-chemical characteristics of both kitchen and bathroom grey water from previous studies.

Parameters	Kitchen Wastewater					Bathroom Wastewater			
	[27]		[28]	[24]	[29]	[30,31]		[28]	[24]
Temperature (°C)				24.4–30.9					25.8–29.0
Total solids (mg/L)			679–1272					777	
Total suspended solids (mg/L)				11–3934	134–1300			58–78	19–793
Total dissolved solids (mg/L)			312–903			810.6			
pH		5.5 ± 0.5	6.5–7.7	5.58–10	5.9–7.4	7.3–7.8	8.1	7.1–7.6	5.98–8.40
BOD (mg/L)			40.8–890	185–2460	536–1460	166 ± 37	7.2	129–173	20–673
COD (mg/L)	770–2050	110 ± 100	58–1340	411–8071	26–2050	575 ± 98	48	230–367	64–903

**Table 2 ijerph-22-00740-t002:** A depiction of some of the heavy metals detected and quantified in grey wastewater globally.

Heavy Metals	Concentrations Quantified Mg/L	Source of Greywater	Country	Citations
Fe	0–0.233	Bathroom	Ghana	[105]
	0–0.15	Shower	India	[106]
	0–0.17	Wash basin	India	[106]
	0–0.81	Kitchen	India	[106]
	0–0.095	Kitchen	Ghana	[105]
	0–1.34	Laundry	India	[106]
	0–0.367	Salon	Ghana	[105]
	0–0.129	Laundry	Ghana	[105]
	0–0.005	Rainwater	India	[107]
Zn	0–0.04–0.10 0–2.59 0–2.03 0–0.11 0–0.055 0–0.018 0–0.044	Domestic Shower India Kitchen Kitchen Laundry Laundry Salon	Ghana India India India Ghana India Ghana India	[105] [106] [106] [106] [105] [106] [105] [107]
Pb	0–0.087	Bathroom	Ghana	[105]
	N.D	Shower	India	[106]
	N.D	Wash basin	India	[106]
	0–0.04	Kitchen	Ghana	[106]
	0–0.095	Kitchen	Ghana	[105]
	<0.03	Laundry	India	[106]
	0–0.099	Laundry	Ghana	[105]
	0.064	Salon	Ghana	[105]
Cd	0–0.001	Bathroom	Ghana	[105]
	0–0.005	Shower	India	[106]
	<0.001	Wash Basin	India	[106]
	N.D	Kitchen	India	[106]
	N.D	Kitchen	Ghana	[105]
	N.D 0.001	Laundry	India	[106]
	0.001	Laundry	Ghana	[105]
	0.001	Salon	Ghana	[105]
Cu	0–0.004	Shower	India	[106]
	<0.005	Wash Basin	India	[106]
	0–0.02	Kitchen	India	[106]
	0–0.10	Laundry	India	[106]
Cr	N.D	Shower	India	[106]
	N.D	Wash Basin	India	
	0–0.39	Kitchen	India	
	<0.004	Laundry	India	

## Data Availability

The data generated in this review.

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
