# Peer review of "Greywater Reuse: Contaminant Profile, Health Implications, and Sustainable Solutions"

_ijerph, 2025, doi:10.3390/ijerph22050740_

Round 1
Reviewer 1 Report
Comments and Suggestions for Authors
In this manuscript, the authors review the water reuse in water-stressed regions. The composition and physicochemical characteristics of greywater were explored. The potential health risks of contaminants in grey water were discussed, which highlighted the importance of safe and sustainable reuse of greywater.
- Some expressions should be consistent throughout the manuscript, such as EDC and EDCs.
- The challenges and trends of water reuse in water-stressed regions should be in-depth discussed in this review.
- The novelty of this review should be highlighted in abstract and introduction compared to similar published reviews.
- Sustainable use of grey water should be reviewed more clearly.
- The removal of contaminants from greywater should be discussed in this review.
Reviewer 2 Report
Comments and Suggestions for Authors
The paper fits to the aim and scope of the journal, however, it has major flaws, which should be revised before publication:
- The title of the paper should be edited in order to emphasis greywater reuse.
- The title refers to water-stress regions, however a review does not reveal what regions are meant and how the water-reuse practices are presented in this regions.
- The abstract should give readers a brief summary of the research. In current version it looks like a pre-introduction
- Line 34. I doubt that dish-washing water may be considered greywater
- The references through-out the entire manuscript should be formatted accoirding to the template
- I suppose, FOG are also chemical parameters of greywater
- subsection 2.2.1 refers to temperature but does not contain any values to describe the ranges of influence. The same repeats in subsection 2.2.2 and futher - what are the variations of pH, TSS, COD, BOD? How does the treatment go in a certain range? The authors refer to many papers, and definetely may present some precise values for each parameter - preferably in a table form. Moreover, which are the typical values for water-stress regions?
- Title of section 3 - "..... grey wastewater". I would recommend to use a unified word "greywater" within the intire text
- The more I read the paper, the more philosophical it became. No values of parameters, which are crucial for any kind of (waste)water treatment. Only general words, how the pollution appear and why it is bad.
- For instance, Line 344. "Elevated levels of copper and zinc...' - what is the range of these elevated levels?
- Finally, authors present table 1. However, it only refers to research India, Ghana and Thailand. I don't think these values are representative enough. Moreover, rainwater was also consiedered as greywater
- In general, the article lacks specific data and values of key pollution indicators of greywater
- The paper would look much better if authors included the best available practices of greywater treatment
In current version, the paper is too general and looks poorly organized. However, I would recommend resubmission after careful revision.
Reviewer 3 Report
Comments and Suggestions for Authors
The paper describes in detail the typical contaminants found in greywater, classified by their sources, potential environmental and health impacts, and recommended treatments for reuse. The manuscript is clearly written and well-structured. However, I cannot recommend the publication in its current form for several reasons.
First, the authors frequently reference studies and data related to wastewater in general rather than strictly focusing on greywater. This is why contaminants from agricultural or industrial or mixed wastewater sources appear several times in the paper, which is not the scope of greywater-specific reuse.
Additionally, the study does not include emerging contaminants such as nanoparticles and nanoplastics, a hot topic in greywater (and in wastewater) nowadays.
Also, although the manuscript provides extensive descriptive content, it lacks a general and critical final discussion. Section 5 (Benefits and Challenges) requires a deep analysis that critically evaluates the implications, limitations, and future perspectives of greywater treatment and reuse.
There is also a significant inconsistency regarding the title and the manuscript content. The title refers in general to water reuse, without specifying greywater explicitly, yet the discussion around benefits and challenges mainly focuses on the impact of untreated greywater. This is contradictory because the main sections of the paper describe suitable treatment options for each kind of contaminant.
In addition, the emphasis in the title on water-stressed regions is unnecessary from my point of view. It is clear that water reuse becomes key in such climates, but decentralized treatment and reuse of greywater (especially for toilet flushing within residential buildings rather than irrigation, which is the main message of the manuscript) is a sustainable and significant strategy to improve urban water management world-wide.
Finally, the figures can be clearly improved. First, an additional figure summarizing suitable treatment methods and traibs for different contaminants according to their potential reuse applications. Second, figures 1 and 2 are unnecessary, as they extensively detail emerging contaminants and endocrine-disrupting compounds (EDCs) primarily associated with industrial processes or excreted via urine, thus not typically present in greywater. A table summarizing the main pollytants and ranges in greywater would be very helpful too.
Round 2
Reviewer 2 Report
Comments and Suggestions for Authors
After revision, the paper has been improved and in current version may be recommended for publication
Reviewer 3 Report
Comments and Suggestions for Authors
I would like to thank the authors for their effort in revising the manuscript. The new version includes important improvements and the paper can be accepted with minor changes. It is a good and complete review of the different types of greywater contaminants, their potential impacts, and the treatment options. The addition of the section on nanoparticles and microplastics is very relevant and clearly improves the manuscript.
On the other hand, I noticed that some words referring to industrial and agricultural contaminants have been removed (as suggested by the reviewers), even if they were originally cited from the scientific references. I am not sure this is very ethical. Also, the rebuttal letter is not very detailed and does not clearly explain all the changes made in the manuscript, but it is ok.
Before final acceptance, I suggest the following minor changes:
-
In the abstract, the sentence "Finally, the review discussed the review highlights the removal of contaminants" is not correct. Please choose one: either "the review discussed" or "the review highlights".
-
Figure 1 has been removed, so Figure 2 should be renumbered as Figure 1.
-
The conclusions are too general. It would be better to include a summary of the main review findings (as they are included in the abstract).
